# Reversing Decline in Aging Muscles: Expected Trends, Impacts and Remedies

**DOI:** 10.3390/jfmk10010029

**Published:** 2025-01-11

**Authors:** Matthew Halma, Paul Marik, Joseph Varon, Jack Tuszynski

**Affiliations:** 1Open Source Medicine OÜ, 6-15 13517 Talinn, Estonia; mhalma@theflccc.org; 2Frontline COVID-19 Critical Care Alliance, Washington, DC 20036, USA; 3Department of Physics, University of Alberta, Edmonton, AB T6G 2M9, Canada; 4Politecnico di Torino, 10129 Torino, Italy

**Keywords:** healthy aging, physical performance tests, cognitive function, metabolic health, longevity interventions, cardiovascular fitness, musculoskeletal strength, quality of life in older people

## Abstract

**Background**: Age-related decline in musculoskeletal function is a significant concern, particularly in Western countries facing demographic shifts and increased healthcare demands. This review examines the typical trajectories of musculoskeletal deterioration with age and evaluates the effectiveness of various interventions in preventing or reversing these changes. **Methods**: The review analyzes documented rates of decline across multiple parameters, including muscle mass, Type II muscle fiber reduction, and decreased motor unit firing rates. It examines evidence from studies on targeted interventions aimed at reversing these trends or preventing further decline. **Results**: The evidence suggests that multimodal interventions, including strength training can effectively maintain or improve physical function in aging adults. These interventions have shown potential in altering the trajectory of age-related decline in musculoskeletal function. Conclusions. The findings of this review have important implications for healthcare providers and policymakers in addressing the challenges of an aging population. By providing a framework for understanding and addressing age-related physical decline through evidence-based interventions, this review offers potential strategies for reducing healthcare costs and improving the quality of life for older adults.

## 1. Introduction

Aging typically is associated with decreases in musculoskeletal function, which results in greater degrees of physical disability, injury risk, and is associated with increased mortality [1]. Presently, Western countries are composed of a large proportion of older adults, who will be entering retirement. The economic implications of the upward shift in the average age is a commonly discussed issue [2,3,4], especially given that healthcare spending is heavily skewed towards end-of-life care [5]. For greater independence and health in the older population, it pays dividends to start interventions early, including physical and cognitive interventions. This review focuses on the musculoskeletal parameters which typically worsen with age, and how the magnitude of the expected decline compares with what can be achieved through modest interventions.

## 2. The Effect of Aging on Physical Capacity

Age-related declines in physical functioning are thought to be quite ubiquitous. Age-related decline in muscle mass occurs throughout the lifespan, at 0.37% per year in women and 0.47% per year in men [6]. This loss occurs mostly in Type II fast-twitch fibers, while Type I fibers are mostly maintained [7]. Protein synthesis rates also decrease with age [8]. Besides the important loss in strength, muscle mass also influences metabolism, and lower muscle mass has negative effects on metabolism.

The skeletal system undergoes parallel deterioration, with decreasing bone mineral density and structural changes in the tendons and connective tissues [9] (see Figure 1). The tendons become less stiff and show decreased mechanical properties, affecting their ability to transfer force effectively from muscles to bones.

The expected trends in the musculoskeletal system with aging are shown in Table 1. These trends show a gradual decline in the functional and structural parameters of muscle, bone, and tendon as aging progresses [9]. In addition, a limited number of these parameters are predictive of the lifespan, which suggests that they are associated with overall health.

For healthy aging, changes made early on can have significant impacts later on in life, especially if they are long-term changes in habits and practices, instead of short-term changes. Physical fitness, including musculoskeletal strength and integrity, is important in old age, as this enables older people to retain independence for longer, to have better overall health, to experience fewer accidents, and to recover quickly from injuries [10].

Falls are one example where physical fitness can mean the difference between life and death. The mortality after hip fractures in older adults varies from approximately 10% [11] up to about 35% [12,13,14]. Fitness can improve the dynamic balance and lower limb strength, which allows people to maintain balance in potential fall scenarios [15].

**Table 1 jfmk-10-00029-t001:** Trends in musculoskeletal parameters with increasing age and corresponding interventions for improving musculoskeletal parameters.

System	Component	Changes	Rate/Magnitude	Association with Healthspan/QOL Measures	Intervention	Magnitude of Change	Mechanism of Adaptation
**Muscular System**	Muscle Mass	Overall muscle mass decline	Women: 0.37%/year Men: 0.47%/year Over 75 years: Women: 0.64–0.70%/year Men: 0.80–0.98%/year [6]	Correlation between thigh muscle area and telomere length [16] Association of appendicular lean mass normalized to body mass index and 10-year health-related quality of life [17] Muscle mass not associated with overall quality of life [18,19] Muscle mass correlated with physical vitality, emotional functioning, and physical functioning in older breast cancer survivors [20]	Strength training (2× per week for 24 weeks)	3.8% [1.6%, 6.1%] increase in bone mineral-free lean tissue mass [21]	Activation of anabolic pathways [22] Satellite cell activation [23] Hormonal response [24]
	Muscle Fibers	Type II (fast twitch) fiber reduction	10–40% reduction in Type II fiber size [25]	In animal models, Type II muscle fibers help to regulate glucose metabolism [26]	Resistance training 3× per week for 12 weeks	28% increase in area [27]	
		Type I (slow twitch) preservation	Decrease is much slower than for Type II [28]		Resistance training 3× per week for 12 weeks	Non-significant increase in Type I muscle fiber size [27]	
	Muscle Function	Reduced force per unit area		Strong association between grip strength and mortality risk [29] Grip strength associated with level of independence in old age [30,31]	Leg press trained 3× per week for 12 weeks	22% increase in leg press power [32]	
		Reduced motor unit firing	Maximum voluntary contraction (MVC) decreases by ~50 points or 9% per decade [33] Maximum firing rate 30–35% lower in older adults [34]		6 weeks of resistance exercise	Maximal motor unit discharge rates were 49% higher for the older adults [35]	Increase in neural drive from CNS to activate muscle fibers [36]
					6 to 12 weeks of resistance training	Voluntary activation of knee extensors increased 1.8% following resistance training [37]	
		Impaired calcium handling	33% reduction in calcium reuptake [38]	Unknown	Selenium supplementation and training	Improvement in calcium release in older mice [39]	Effect mediated through ryanodine receptor [39] and selenoprotein N [40]
					Acute exercise training	Improvements in calcium release rate [41]	Modification of calcitropic hormone levels [42]
					12 weeks of high-resistance strength training	Partial reversal of reduction in sarcoplasmic reticulum Ca ^2+^ uptake in skeletal muscle [38].	
	Muscle Quality	Increased fat infiltration	Non-contractile area in leg anterior compartment approximately 2.5-fold larger in older subjects than in young subjects [43,44]	Arm fat mass index association with increased non-cardiovascular mortality [45] Low fat mass and high muscle mass associated with a 62% [32%,78%] decrease in total mortality [46] Skeletal muscle fat infiltration associated with higher all-cause and cardiovascular mortality [47]	12-week resistance training program	11% decrease in thigh intramuscular adipose tissue [48]	Increases fatty acid oxidation [49] Improves insulin sensitivity [50] Reduces adipogenic signaling [51]
					High-effort single-set exercise training 2×/week for 16 months	Fat infiltration stable in exercise group but increased in control group [52]	
		Greater fibrosis	Increase in tissue fibrosis observed in aging mouse model [53] Increase in muscle fibrosis biomarkers in humans [54]		Metformin	Lowered biomarkers of muscle fibrosis [55]	Inhibits TGF-β1 signaling, alters fate of myofibroblasts [56]
					Nintedanib	Decreased expression of fibrotic genes [57]	Reduces proliferation and migration of fibroblasts [57], downregulation of extracellular matrix production [57], interference with profibrotic signaling [58]
					Resistance training and Dioscorea esculenta	Circulating levels of C1q, a biomarker associated with fibrosis, lower in experimental group [59]	Lowers C1q [59]
					Losartan	Decreases fibrosis and fibrotic biomarkers in animal models [60]	Interferes with TGF-β1 signaling [61]
					Suramin	Decreases fibrosis in animal models [60]	Reduces TGF-β signaling [62]
					Decorin	Decreases fibrosis in animal models [60]	Inhibits TGF-β [63]
					Halofuginone	Decreases fibrosis in animal models [60]	Inhibits collagen synthesis [64]
		Reduced sensitivity to anabolic stimuli	Decreased muscle protein synthesis after insulin infusion in older people compared to younger people [65]	N/A	Leucine	Leucine supplementation stimulates muscle protein synthesis [66]	Potent stimulator of mTORC1 and protein synthesis [67]
					High protein intake	Doubling the recommended daily intake of protein increased muscle protein synthesis by 19% [68]	Increasing muscle protein synthesis [69]
	Pennation angle	Decrease in pennation angle	4% decline per decade in vastus lateralis pennation angle [70]	No direct correlation, but pennation angle is correlated with age with r = −0.50 [70]	Leg press training for 10 weeks	30% increase in pennation angle of vastus lateralis (VL) muscle [71]	Remodeling [72]
	Mitochondrial number	Decrease with age	Older participants (71+/−2 years) have 57% reduction in mitochondria compared to younger individuals (23+/−2 years) or 1.74% reduction per year. [73]	Association with VO2 max [74]	30 to 60 min 3× per week for 4 months at moderate intensity (75% of maximum heartrate)	Increase in muscle mitochondrial density by 50.7% in exercise group [75]	Increases muscle GLUT−4 levels and insulin action [76,77]
	Energy expenditure at rest	Decreased basal metabolic rate	4% decline per decade after 50 years of age [53,78]	Genetically predicted basal metabolic rate negatively associated with lifespan [79]	Fish oil with resistance training	Significant increase in resting metabolic rate [80]	Increases in the phosphorylation status of kinases related to the mTORC-1 signaling, increased muscle mass [81]
					Nutritional consultation and physical training	Increase in RMR [82]	Maximizes muscle protein synthesis (MPS) through the activation of mammalian target of rapamycin (mTOR) [83]
**Skeletal System**	Bone Structure	Accelerated bone mineral density loss	Cortical zone in upper femoral neck declined by 6.4% per decade [84]	One standard deviation difference in bone mineral density is associated with a 1.39-fold increase in mortality [85] 1 SD increase in total hip BMD associated with a 0.77 [0.61,0.91] relative risk of overall mortality [86]	Physical exercise program	No decrease in bone mineral density of exercise group, decrease of 1.1% [0.1%,2.1%] in control group [87]	Activating mTORC1 and Mitogen-active protein kinases signaling for growth, repair, and adaptation [88]
		Elastic modulus	2.3% reduction per decade of life past age 35 [89]	Unknown	N/A	N/A	
		Bone strength	Reduction by 3.7% each decade after age 35 [89]	Unknown	Physical exercise	Reduction in fracture risk by 51% in the exercise group [90]	
		Fracture toughness	Kc: 4.1% reduction per decade after age 35 [89]	Unknown			
		Work of fracture	8.7% reduction per decade after age 35 [89]	Unknown			
	Tendons	Reduced stiffness	Decreased maximum shortening velocity of tendons between older (75 years) and younger (20 years) adults by 16% [91]	Unknown	14 weeks of high-load resistance training	Increase in tendon stiffness by 65% [92,93]	Tendon hypertrophy [94,95]
		Decreased shock absorption	F20 of occiput (inverse proxy for shock absorption capacity) roughly doubles between third and fifth decade of life, and between the fifth and seventh decade of life [96]	Decreased shock absorption capacity may contribute to greater risk from falls [97]	14 weeks of high-load resistance training	Tendon Young’s modulus increased by 69% [92,93]	
	Function	Flexibility	5–6 degree decline per decade in shoulder abduction [98]	Not associated with mortality [99].	Dynamic and static stretching exercises for 12 weeks	Sit and reach test improvement by 23 +/−10% [100]	Stretching promotes the addition of sarcomeres in series, increasing the functional length of muscle fibers and improving the ability to generate an extended ROM [101], improved muscle–tendon unit (MTU) compliance [102,103]

## 3. Interventions for Improving the Aging Musculoskeletal System

Resistance training produces many important benefits for aging bodies, across a wide degree of systems, organs, and tissue types. Some training specificity may be necessary to target certain parameters, though the benefits of resistance training manifest across a wide variety of training regimens. Table 1 includes interventions that can improve the musculoskeletal parameters that secularly decline with age, and the magnitude of their impact.

In Table 2, we include four aspects of fitness that are helpful for older people and where the secular trend declines with aging. These may also be assessed using either biomarker- or function-based tests, and training interventions prescribed accordingly.

Adaptation to interventions (including physical training) is characterized by a wide range of molecular mechanisms (Table 1). For certain parameters like strength, the value can be improved through multiple mechanisms, including nervous system adaptations as well as hypertrophy of the relevant muscles [22]. Other aspects of fitness which this article has only briefly covered include endurance training, balance training, and flexibility. Endurance training can be important for developing cardiorespiratory fitness, which is highly associated with lifespan [111]. Balance training is important in the context of fall prevention, which can carry a significant mortality risk for older people [112]. Flexibility is associated with functional ability in adults [113]. Overall, the fitness level is associated with greater independence and capability into old age.

## 4. Conclusions

This review demonstrates how the trajectory of age-related physical decline can be improved through targeted interventions. Research shows meaningful improvements across various physiological parameters when using multiple strategies, from resistance training to nutritional optimization. Physical exercise has demonstrated remarkable effects on both muscular and skeletal health, while proper nutrition supports muscle synthesis and overall function.

The integration of functional and biomarker testing with personalized interventions offers a practical framework for promoting healthy aging. This approach is particularly relevant given the growing demographic of older adults in Western societies and their associated healthcare challenges. Looking ahead, there is a clear need for continued research to refine these interventions and explore their synergistic effects.

Healthcare providers and policymakers can leverage these insights to test adults for their physical capacities and prescribe training programs to bring up the parameter values, which will result in greater health outcomes. People are more likely to follow an individualized program than a generic program, more so than vague advice to ‘eat right’ and ‘move more’ [114,115]. These training programs can be used with a wearable device, which allows for gamification, accountability, and allows for the monitoring of progress [116], as well as the development of more effective programs supporting healthy aging, potentially reducing the burden of age-related chronic diseases and associated healthcare costs. Through the systematic application of these findings, we can work toward a future where aging is characterized by maintained functionality and enhanced quality of life rather than decline.

While this review has focused mostly on physical interventions, other interventions, such as nutritional supplementation and novel therapeutics practices, may also be of interest, especially as they may reduce the time taken to engage in a program designed to decrease the rate of decline in musculoskeletal function with age. The potential limitations of this approach are that older people may not have access to a gym, but many exercise interventions can be performed with minimal or no equipment. Furthermore, this study relies on the findings in the cited studies being robust, which is a challenge inherent to providing guidelines based on clinical research.

Additionally, examining the impact of interventions and the association between musculoskeletal parameters and healthspan or, inversely, mortality. Each intervention creates a change in the expected trend for an aging person, which can improve health outcomes. This can further be extrapolated to financial costs and may alleviate the cost of old-age care.

## Figures and Tables

**Figure 1 jfmk-10-00029-f001:**
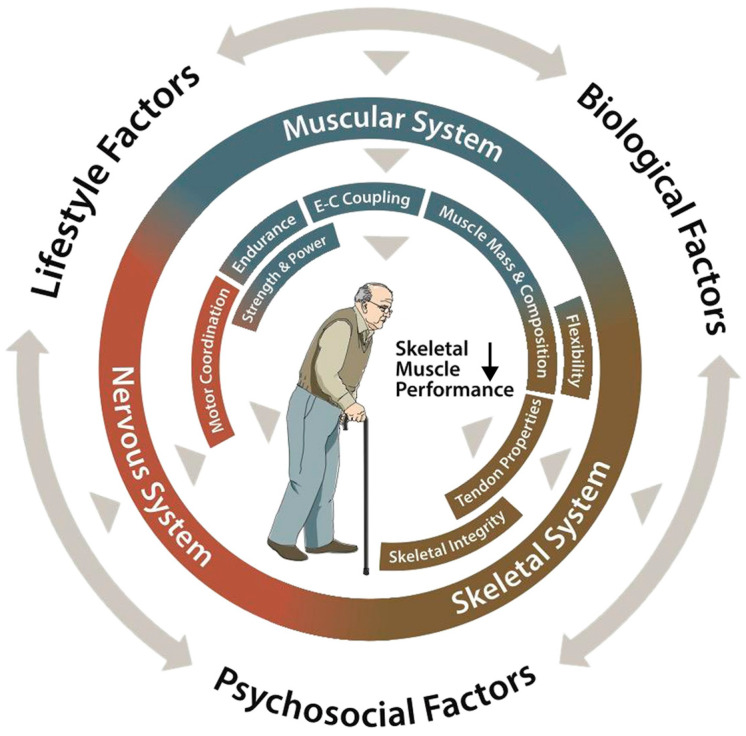
Declining skeletal muscle performance in older people. Diagram by Tim Goheen, Associate Professor, Ohio University School of Visual Communication. Reproduced from [9] under a CC BY-NC 4.0 license (https://creativecommons.org/licenses/by-nc/4.0/, accessed on December 10, 2024).

**Table 2 jfmk-10-00029-t002:** Intervention classes for maintaining physical function with age and means of training.

Training Type	Trend (Absent Training)	System	Associated Tests	Training	Adaptations
Strength Training	Sarcopenia, muscle loss, bone loss	Musculoskeletal	Grip strength [104]	Weightlifting	Increase in muscle mass and bone density
Endurance training	Lower VO2 max	Metabolic, cardiopulmonary	Resting metabolic rate, creatine phosphokinase [105]	Running, swimming, walking, cycling, cross-country skiing, hiking, etc.	Increased mitochondrial size, greater ability to metabolize fat, increased (heart) stroke volume
Balance training	Poorer coordination	Musculoskeletal, nervous	Self-selected gait velocity [106], chair rise test (timed 5 chair rises), tandem standing and walking, timed up and go test, clinical gait analysis with special focus on regularity, mechanography [107]	Yoga	Neuromuscular control [108]
Flexibility	Decrease in joint flexion [98,109]	Musculoskeletal, tendons, fascia	Flexibility tests: Flexindex [109]	Yoga, pilates	Improved flexibility and stability [110]

## Data Availability

No new data were created or analyzed in this study. Data sharing is not applicable to this article.

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
