# Peer review of "Reversing Decline in Aging Muscles: Expected Trends, Impacts and Remedies"

_jfmk, 2025, doi:10.3390/jfmk10010029_

Round 1

Reviewer 1 Report

Comments and Suggestions for Authors

Thank you for submitting this manuscript. The overall aim is clear to see but I think the manuscript could benefit from more structure. I am not sure the authors are using biomarkers in the correct sense in this manuscript so suggest using a different descriptor for the measures they have tabulated. Also, with any author writing about older people, one should know by now to refrain from using the word elderly and substitute for older.

The authors need to

1. Justify clearly the aims and objectives of this paper - why is it worthwhile for the readership?

2. Structure the paper so the they effectively disentangle what are physiological changes with age, when physiology becomes pathophysiology and what we can do about these pathophysiological changes to promote health ie the interventions the authors right about. Maybe within a section use subheadings - physiological and pathophysiological changes & interventions. At this present time the manuscript is an accumulation of sentences and short paragraphs as opposed to a story.

3. End  with key points - what are the key points we need to focus on to promote healthy aging and how to we achieve this based on the manuscript?

Author Response

Comment 1: 

Thank you for submitting this manuscript. The overall aim is clear to see but I think the manuscript could benefit from more structure. I am not sure the authors are using biomarkers in the correct sense in this manuscript so suggest using a different descriptor for the measures they have tabulated. Also, with any author writing about older people, one should know by now to refrain from using the word elderly and substitute for older.

Response 1: We have changed the use of the term biomarkers to physical measurements, except in two cases where a biomarker measurement is used in lieu of the quantity of interest and a mention of incorporating biomarkers into a broader aging panel in the discussion.

Comment 2: 

The authors need to

1. Justify clearly the aims and objectives of this paper - why is it worthwhile for the readership?

2. Structure the paper so the they effectively disentangle what are physiological changes with age, when physiology becomes pathophysiology and what we can do about these pathophysiological changes to promote health ie the interventions the authors right about. Maybe within a section use subheadings - physiological and pathophysiological changes & interventions. At this present time the manuscript is an accumulation of sentences and short paragraphs as opposed to a story.

Response 2: Thank you for your comment, indeed the structure of the previous manuscript could be improved. We have reduced our topic to physical changes in older people, focusing mainly on musculoskeletal system changes by parameter. We have structured the flow as presenting a particular parameter and the trend that occurs with increasing chronological age, along with any known association between that parameter and lifespan, mortality or quality of life variables. We then introduce the interventions for that particular parameter and include the magnitude of the change expected by following the interventions, typically a resistance training program. 

Comment 3: 3. End  with key points - what are the key points we need to focus on to promote healthy aging and how to we achieve this based on the manuscript?

Response 3: Thank you for your comment. We have updated our conclusion section to emphasize a) trends in musculoskeletal parameters can be alleviated or even reversed through interventions including exercise with adequate protein intake. With this, given the scale of aging, it is important to test and intervene early as many of these measures are associated with greater quality of life and healthspan. 

Reviewer 2 Report

Comments and Suggestions for Authors

The authors present a review article titled: “Life in the Years: Maximizing Physical and Cognitive Health in the Elderly”.

I have carefully reviewed the manuscript and would like to provide the following feedback.

-        While reviewing the tables, I noticed several abbreviations and terms that would benefit from further clarification. For instance, Table 1 mentions "DunedinPoAm, PhenoAge, and GrimAge clocks," and Table 3 references "BDNF." Including descriptions of these terms would be helpful for the reader.

-        The sentence in line 157 appears to be incomplete. I would recommend a critical review of this section to ensure clarity and accuracy.

-        The information and associations presented in the study, in my opinion, do not offer groundbreaking new insights. Specifically, the conclusions section (spanning only 23 lines, line 221-44) is underrepresented in the manuscript and primarily reiterates well-known aspects. For example, the suggestion that addressing musculoskeletal strength, cognition, cardiovascular and pulmonary health as well as emotional well-being, is beneficial for healthier ageing, seems relatively self-evident.

-        Given the content of the review, it raises the question of what individual actions can contribute to healthier aging. For example, considering the cardiovascular risk factors and the influence of physical activity on healthy aging, an interesting addition could be a discussion on whether the use of smartwatches with health monitoring features (e.g., heart rate, and oxygen levels and step tracking) might aid in the early detection of risk factors and facilitate timely interventions.

-         In summary, I recommend that the authors explicitly outline how their study advances current knowledge and explore practical approaches with potential benefits for older individuals.

Author Response

Comment 1: While reviewing the tables, I noticed several abbreviations and terms that would benefit from further clarification. For instance, Table 1 mentions "DunedinPoAm, PhenoAge, and GrimAge clocks," and Table 3 references "BDNF." Including descriptions of these terms would be helpful for the reader.

Response 1: We thank the reviewer for this point and believe it would constitute an improvement. However, we have restructured this manuscript to remove this table and the section of the manuscript focusing on cognitive health, instead tailoring it for muscular changes in older people and their implications. 

Comment 2: -        The sentence in line 157 appears to be incomplete. I would recommend a critical review of this section to ensure clarity and accuracy.

Response 2: 

We thank the reviewer for noticing this. We have removed the improper sentence, as it’s meaning is conveyed better in the previous sentence: “Endurance exercise proves particularly beneficial for aging individuals. It improves mitochondrial density through the enlargement of existing mitochondria [105–107]and these benefits are maintained over time.”

Comment 3: The information and associations presented in the study, in my opinion, do not offer groundbreaking new insights. Specifically, the conclusions section (spanning only 23 lines, line 221-44) is underrepresented in the manuscript and primarily reiterates well-known aspects. For example, the suggestion that addressing musculoskeletal strength, cognition, cardiovascular and pulmonary health as well as emotional well-being, is beneficial for healthier ageing, seems relatively self-evident.

Response 3: We agree with the reviewer that it is not groundbreaking to emphasize the impact of exercise on well-being in the elderly. We have tailored it towards a focus on various parameters of musculoskeletal health and the trends which occur with aging for each parameter. These parameters are then connected with interventions which can reverse or halt the process that occurs with aging. We believe future work on this can build an averted cost model, using the data on the impact of the intervention, and the predictive power of the parameter for healthspan, lifespan or mortality. 

Comment 4: Given the content of the review, it raises the question of what individual actions can contribute to healthier aging. For example, considering the cardiovascular risk factors and the influence of physical activity on healthy aging, an interesting addition could be a discussion on whether the use of smartwatches with health monitoring features (e.g., heart rate, and oxygen levels and step tracking) might aid in the early detection of risk factors and facilitate timely interventions.

Response 4: We thank the reviewer for his or her comment. This is very useful to note as wearable technology can improve adherence to programs, in addition to apps and personalized digital programs. We have added the following sentence to our conclusion.

"Healthcare providers and policymakers can leverage these insights to test adults for their physical capacities and prescribe training programs to bring up the parameter values, which will result in greater health outcomes. People are more likely to follow an individualized program than a generic program, moreso than vague advice to ‘eat right’ and ‘move more’ [132,133]. These training programs can be used with a wearable, which allows for gamification, accountability, and allows for monitoring of progress [134]."

Comment 5:  In summary, I recommend that the authors explicitly outline how their study advances current knowledge and explore practical approaches with potential benefits for older individuals.

Response 5: We thank the reviewer for his or her comments. We have altered the structure to focus on musculoskeletal parameters and we advance current knowledge by providing an overview of the means to remedy expected (negative) trends in musculoskeletal parameters with increasing age. We are novel in providing the nucleus for an averted cost model through early intervention in the musculoskeletal function of adults. 

Round 2

Reviewer 1 Report

Comments and Suggestions for Authors

Thank you for re submitting the manuscript.

1. I found the manuscript still difficult to read as it still comprises a collection of unrelated statements put together. The manuscript, I am afraid, still lacks the structure, specificity and  flow that would be expected of a review article. 

2. There are inaccuracies - age related decline is an inevitable process - this is consequent to physiology of aging. We are able to modify the trajectory through interventions.

3. The loss of muscle mass is not solely due to loss of type II fibers, it is a combination and atrophy of both type I & II. I appreciate literature support preferential loss of type II. Authors have not mentioned denervation.

3. Table 1&2, although contain good information are too long and non specific. Choose what parameter is most clinically relevant and show what interventions are beneficial - this would make the information more appealing to a wider audience. Suggest combine tables 1&2.

4. Remove elderly form Figure 1

5. What does metabolically unwell mean - this has no meaning in the clinical sense.

6. I am unclear what the metabolism and endurance sections are trying to say as the authors switch from one statement to the next and introduce some quite specific illnesses to illustrate points - i.e, breast cancer, CFS. What about gathering evidence from aging individuals - what does the literature say about reversing/intervening on metabolic perturbations or lack of physical activity in older people - the topic of this review.

Author Response

Thank you for re submitting the manuscript.

  1. I found the manuscript still difficult to read as it still comprises a collection of unrelated statements put together. The manuscript, I am afraid, still lacks the structure, specificity and  flow that would be expected of a review article. 

We have made efforts to streamline the flow of the article by removing extraneous detail and focus on the adaptations of the musculoskeletal system to aging and to targeted interventions like physical exercise, then transitioning to the potential impact early intervention can have for the health of older individuals.

  1. There are inaccuracies - age related decline is an inevitable process - this is consequent to physiology of aging. We are able to modify the trajectory through interventions.

We have removed or altered our statements to communicate the more conservative phrasing shared by the reviewer, that the trajectory of age-related decline may be improved through targeted interventions.

  1. The loss of muscle mass is not solely due to loss of type II fibers, it is a combination and atrophy of both type I & II. I appreciate literature support preferential loss of type II. Authors have not mentioned denervation.

We have elaborated that the rate of decline for Type I is less than Type II, and not necessarily zero. We have discussed the nervous system adaptations to both aging as well as to exercise through our inclusion  of motor unit firing in the main table

4.  Table 1&2, although contain good information are too long and non specific. Choose what parameter is most clinically relevant and show what interventions are beneficial - this would make the information more appealing to a wider audience. Suggest combine tables 1&2.

It is a good idea to combine the tables. We have added two extra columns to Table 1 to include the interventions from Table 2 and their effects on the parameter.

5.Remove elderly form Figure 1

We have removed the word ‘elderly’ from this figure.

6.What does metabolically unwell mean - this has no meaning in the clinical sense.

We have removed these imprecise and vague statements from the manuscript.

7. I am unclear what the metabolism and endurance sections are trying to say as the authors switch from one statement to the next and introduce some quite specific illnesses to illustrate points - i.e, breast cancer, CFS. What about gathering evidence from aging individuals - what does the literature say about reversing/intervening on metabolic perturbations or lack of physical activity in older people - the topic of this review.

We have removed these sections instead focusing on the interventions working on the musculoskeletal system in old age.

Reviewer 2 Report

Comments and Suggestions for Authors

The authors have fundamentally revised and improved their manuscript. The focus of the study is now on musculoskeletal health, and its title changed to: “Reversing decline in aging muscles: Expected trends, impacts and remedies”.

The adjustments to the tables provide a very informative overview. I have two minor points that should still be addressed:

-        The title of Table 2 does not align with its new contents and should be adjusted accordingly.

-        I recommend presenting the limitations of the study more clearly and precisely.

Author Response

he authors have fundamentally revised and improved their manuscript. The focus of the study is now on musculoskeletal health, and its title changed to: “Reversing decline in aging muscles: Expected trends, impacts and remedies”.

The adjustments to the tables provide a very informative overview. I have two minor points that should still be addressed:

-        The title of Table 2 does not align with its new contents and should be adjusted accordingly.

We have merged tables 1 and 2 into Table 1. Table 1’s caption now reads “Trends in musculoskeletal parameters with increasing age and corresponding interventions for improving musculoskeletal parameters.”

-        I recommend presenting the limitations of the study more clearly and precisely.

We have added the following to express study limitations

“While this review has focused mostly on physical interventions, other interventions, such as nutritional supplementation and novel therapeutics practices, may also be of interest, especially as they may reduce the time taken to engage in an anti-aging pro-gram. Potential limitations of this approach are that older people may not have access to a gym, but many exercise interventions can be performed with minimal or no equipment. Furthermore, this study relies on the findings in the cited studies being robust, which is a chellenge inherent to providing guidelines based on clinical research.”

Round 3

Reviewer 1 Report

Comments and Suggestions for Authors

The authors have made the manuscript more succinct and is publishable.A few words in the discussion must be changed before publication:

"and novel therapeutics practices, may also be of interest, especially as they may reduce the time taken to engage in an anti-aging...."

There are no viable novel therapeutic practices, no drugs to reverse MSK ageing or aging per se. Please modify.

Anti-ageing practices is the grail of geriatric research  -  there are no such anti ageing practices across the globe today. These terms are misleading and must be modified in the discussion.

Author Response

Comments1 : 

The authors have made the manuscript more succinct and is publishable.A few words in the discussion must be changed before publication:

"and novel therapeutics practices, may also be of interest, especially as they may reduce the time taken to engage in an anti-aging...."

There are no viable novel therapeutic practices, no drugs to reverse MSK ageing or aging per se. Please modify.

Anti-ageing practices is the grail of geriatric research  -  there are no such anti ageing practices across the globe today. These terms are misleading and must be modified in the discussion.

Response: This is an important clarification and words like anti-aging, while common in the lay lexicon, are not accurate. We have changed the above wording to: "... and novel therapeutics practices, may also be of interest, especially as they may reduce the time taken to engage in an program designed to decrease the rate of decline in musculoskeletal function with age"

We thank Reviewer 1.